# The Renal Clearable Magnetic Resonance Imaging Contrast Agents: State of the Art and Recent Advances

**DOI:** 10.3390/molecules25215072

**Published:** 2020-11-01

**Authors:** Xiaodong Li, Yanhong Sun, Lina Ma, Guifeng Liu, Zhenxin Wang

**Affiliations:** 1Department of Radiology, China-Japan Union Hospital of Jilin University, Xiantai Street, Changchun 130033, China; xiaodong20@mails.jlu.edu.cn; 2State Key Laboratory of Electroanalytical Chemistry, Changchun Institute of Applied Chemistry, Chinese Academy of Sciences, Changchun 130022, China; yhyan@ciac.ac.cn (Y.S.); malina@ciac.ac.cn (L.M.)

**Keywords:** magnetic resonance imaging contrast agents, renal clearance, nanodots, gadolinium (III)-based composites

## Abstract

The advancements of magnetic resonance imaging contrast agents (MRCAs) are continuously driven by the critical needs for early detection and diagnosis of diseases, especially for cancer, because MRCAs improve diagnostic accuracy significantly. Although hydrophilic gadolinium (III) (Gd^3+^) complex-based MRCAs have achieved great success in clinical practice, the Gd^3+^-complexes have several inherent drawbacks including Gd^3+^ leakage and short blood circulation time, resulting in the potential long-term toxicity and narrow imaging time window, respectively. Nanotechnology offers the possibility for the development of nontoxic MRCAs with an enhanced sensitivity and advanced functionalities, such as magnetic resonance imaging (MRI)-guided synergistic therapy. Herein, we provide an overview of recent successes in the development of renal clearable MRCAs, especially nanodots (NDs, also known as ultrasmall nanoparticles (NPs)) by unique advantages such as high relaxivity, long blood circulation time, good biosafety, and multiple functionalities. It is hoped that this review can provide relatively comprehensive information on the construction of novel MRCAs with promising clinical translation.

## 1. Introduction

Magnetic resonance imaging (MRI) is extensively used as a noninvasive, nonionizing, and radiation-free clinical diagnosis tool for detection and therapeutic response assessment of various diseases including cancer, because it can provide anatomical and functional information of regions-of-interest (ROI) with high spatial resolution through manipulating the resonance of magnetic nucleus (e.g., ^1^H) in the body via an external radiofrequency pulse magnetic field [1,2,3,4,5,6,7]. Although it is possible to generate high-contrast images of soft tissues for diagnosis by manipulating pulse sequences alone, MRI is able to further highlight the anatomic and pathologic features of ROI through utilized in concert with contrast agents. Magnetic resonance imaging contrast agents (MRCAs) play an extremely important role in modern radiology because the growth of contrast-enhanced MRI has been remarkable since 1988 [3,4,5,6,7,8]. Up to date, the commercially approved MRCAs and the MRCAs in clinical trial are shown in Table 1.

In general, there are two imaging modes of MRI, named as T_1_- or T_2_-weighted MRI, which have been employed to acquire the restored or residual magnetization by adjusting parameters in either the longitudinal direction or the transverse plane, respectively. The T_1_-weighted MR image shows bright signal contrast (recovered magnetization), while the T_2_-weighted MR image shows dark signal contrast (residual magnetization). The T_1_- or T_2_-weighted MRI can be performed in one machine by simply adjusting the acquisition sequences during the MRI process. Consequently, the MRCAs are divided into two categories based on their dominant functions in T_1_- or T_2_-weighted MRI. Paramagnetic metal nanomaterials/complexes are usually designated as T_1_-weighted MRCAs, which cause bright contrast in T_1_-weighted MR images [3,4,8,9,10]. For example, gadolinium (III) (Gd^3+^) has seven unpaired electrons and a long electron spin relaxation time, which can efficiently promote the longitudinal ^1^H relaxation. Gd^3+^-diethylenetriamine penta-acetic acid (Gd-DTPA) was synthesized as the first T_1_-weighted MRCA and used for contrast-enhanced T_1_-weighted MRI of intracranial lesions in 1988 [8]. Superparamagnetic nanoparticles (e.g., superparamagnetic iron-oxide nanoparticles (SPIONs)) are normally used as T_2_-weighted MRCAs, which provide dark contrast in MR images [11,12,13,14,15,16,17,18,19]. The advantages and disadvantages of Gd^3+^- and SPION-based MRCAs have been discussed in the reviews which are published elsewhere [3,4,5,6,7,8,9,10,11,12,13,14,15,16,17,18,19]. In particular, the advent of nephrogenic systemic fibrosis (NSF) and bone/brain deposition has led to increased regulatory scrutiny of the safety of commercial Gd^3+^ chelates [9]. Figure 1 shows the typical paramagnetic cations including transition metallic cations and lanthanide cations, which are capable of enhancing contrast on MR images. The cations contain unpaired electrons in 3d electron orbitals (transition metallic cations) and/or 4f electron orbitals (lanthanide cations). In addition, several strategies have been proposed for development of T_1_-/T_2_-weighted dual-mode MRCAs, because T_1_-/T_2_-weighted dual-mode MRI can provide an accurate match of spatial and temporal imaging parameters [12]. Therefore, the accuracy and reliability of disease diagnosis can be clearly improved by synergistically enhancing both T_1_-/T_2_-weighted contrast effects.

Because of their unique physiochemical and magnetic properties, magnetic nanoparticles (MNPs) have attracted considerable attention in the construction of MRCAs with high performance during the last two decades (as shown in Figure 2), and they exhibit high potential for clinical applications in MRI-guided therapy. The synthesis, properties, functionalization strategies, and different application potentials of MNPs have been reviewed in detail in the literature published elsewhere [11,12,13,14,15,16,17,18,19]. For example, the MNPs can be used as excellent MRCAs for sensitive detection of tumors because MNPs can efficiently accumulate in tumor through the leaky vasculature of tumor (also known as the enhanced permeability and retention (EPR) effect). The contrast efficacy and in vivo fate of the MNP-based MRCAs are strongly dependent on their physical and chemical features including shape, size, surface charge, surface coating material, and chemical/colloidal stability. For example, the PEGylated MNPs normally have relatively longer blood circulation time and higher colloidal/chemical stability than those of uncoated MNPs [11]. Very recently, the structure-relaxivity relationships of magnetic nanoparticles for MRI have been summarized in detail by Chen et al. [15]. Among all the characteristics of MNP, size plays a particularly important role in the biodistribution and blood circulation half-life of MNP [15,18,20]. The long blood circulation half-life of MNP can significantly increase the time window of imaging. However, the nonbiodegradable MNPs with large hydrodynamic size (more than 10 nm) exhibit high uptake in the reticuloendothelial system (RES) organs such as lymph nodes, spleen, liver, and lung, which causes slow elimination through hepatobiliary excretion [21]. The phenomenon increases the likelihood of toxicity in vivo [22], and severely hampers translating MNPs into clinical practices because the United States Food and Drug Administration (FDA) requires that any imaging agent (administered into the body) should be completely metabolized/excreted from the body just after their intended medical goals such as image-guided therapy [23]. The conundrum could be sorted out by the development of renal clearable MNPs since renal elimination enables rapidly clear intravenously administered nanoparticles (NPs) from circulation to be excreted from body. Due to the pore size limit of glomerular filtration in the kidney, only the MNPs with small hydrodynamic diameter (less than 10 nm) or biodegradable ability are able to balance the long blood circulation half-life time for imaging and efficient renal elimination for biosafety. In addition, although they have similar small hydrodynamic diameters, the negatively charged or neutral NPs are more difficult to be eliminated by the kidney than their positively charged counterparts.

Herein, this review provides the state of the art of renal clearable composites/MNPs-based MRCAs with particular focus on several typical formats, namely Gd^3+^-complex-based composites and magnetic metal nanodots (MNDs), by using illustrative examples. The MNDs mean MNPs with ultrasmall hydrodynamic size (typically less than 10 nm in diameter) such as ultrasmall Gd_2_O_3_ NPs, ultrasmall NaGdF_4_ NPs, ultrasmall Fe_2_O_3_/Fe_3_O_4_ NPs and ultrasmall polymetallic oxide NPs. The gathered data clearly demonstrate that renal clearable composites/MNDs offer great advantages in MRI, which shows a great impact on the development of theranostic for various diseases, in particular for cancer diagnosis and treatment. In addition, we also discussed current challenges and gave an outlook on potential opportunities in the renal clearable composites/MNDs-based MRCAs.

## 2. Gadolinium (III)-Complex-Based Composites

Up to date, all of the MRCAs commercially available in clinic are small molecule Gd^3+^-complexes, which are used in about 40% of all MRI examinations (i.e., about 40 million administrations of Gd^3+^-complex-based MRCA) [24]. However, the small molecule Gd^3+^-complexes generally have short blood circulation time with a typical elimination half-life of about 1.5 h [8,9,25,26,27]. The rapid clearance characteristic of small molecule Gd^3+^-complex makes it difficult to conduct the time-dependent MRI studies. Conjugation of Gd^3+^-complexes with biodegradable materials (e.g., polymers) and/or renal clearable NDs can not only significantly prolong blood circulation of Gd^3+^-complexes, but also improve accumulation amount of Gd^3+^-complexes in solid tumors through an EPR effect [28,29,30,31,32,33,34,35,36,37,38,39,40,41,42,43,44,45,46,47,48,49,50,51,52]. Therefore, the Gd^3+^-complex-based composites enable one to provide an imaging window of a few hours before it is cleared from the body, and additionally enhance the resulting MR signal of a tumor. In particular, biodegradability or small size that ensures the complete clearance of Gd^3+^-complex-based composites within a relatively short time (i.e., a day) by renal elimination after diagnostic scans.

As early as 2012, Li and coauthors synthesized a poly(*N*-hydroxypropyl-l-glutamine)-DTPA-Gd (PHPG-DTPA-Gd) composite through conjugation of Gd-DTPA on the poly(*N*-hydroxypropyl-l-glutamine) (PHPG) backbone [34]. The longitudinal relaxivity (r_1_) of PHPG-DTPA-Gd (15.72 mM^−1^⋅S^−1^) is 3.7 times higher than that of DTPA-Gd (Magnevist^T^). PHPG-DTPA-Gd has excellent blood pool activity. The in vivo MRI of C6 glioblastom-bearing nude mouse exhibited significant enhancement of the tumor periphery after administration of PHPG-DTPA-Gd at a dose of 0.04 mmol Gd kg^−1^ via tail vein, and the mouse brain angiography was clearly delineated up to 2 h after injection of PHPG-DTPA-Gd. Degradation of PHPG-DTPA-Gd by lysosomal enzymes and hydrolysis of side chains lead to complete clearance of Gd-DTPA moieties from the body within 24 h though the renal route. Shi and coauthors synthesized a Gd-chelated poly(propylene imine) dendrimer composite (PPI-MAL DS-DOTA(Gd)) through chelation of Gd^3+^ with tetraazacyclododecane tetraacetic acid (DOTA) modified fourth generation poly(propylene imine) (PPI) glycodendrimers (as shown in Figure 3a) [36]. The r_1_ of PPI-MAL DS-DOTA(Gd) is 10.2 mM^−1^⋅s^−1^, which is 3.0 times higher than that of DOTA(Gd) (3.4 mM^−1^⋅s^−1^). The as-synthesized PPI-MAL DS-DOTA(Gd) can be used as an efficient MRCA for enhanced MRI of blood pool (aorta/renal artery, as shown in Figure 3b) and organs in vivo. The PPI-MAL DS-DOTA(Gd) exhibits good cytocompatibility and hemocompatibility, which can be metabolized and cleared out of the body at 48 h post-injection.

In comparison with small molecule Gd^3+^-complexes, Gd^3+^-complex-based nanocomposites provide significant advantages for contrast-enhanced MRI such as increased Gd^3+^ payload, prolonged blood circulation, enhanced r_1_ and improved uptake of Gd^3+^ [28,29,30]. Generally, the contrast efficiency of Gd^3+^-complex-based nanocomposite is affected by the structure and surface chemistry of the used nanomaterial. For example, the MRI contrast capability of Gd^3+^-complex-metal-organic framework (MOF) nanocomposite is strongly dependent on the size and pore shape of MOF [38]. Furthermore, the nanocomposites of Gd^3+^-complexes with NDs not only have relatively long blood circulation time, but also integrate the properties of both ND and Gd^3+^-complexes [37,38,39,40,41,42,43,44,45,46,47,48,49,50,51,52]. The relatively easy modification property of ND offers opportunities for generation of theranostics for various biomedical applications such as targeted delivery, multimodal imaging, and imaging-guided therapy. For instance, coupling of Gd^3+^-complexes with optical NDs (e.g., quantum dots (QDs), noble metal nanoclusters, and carbon NDs) we can generate contrast agents for MR/fluorescence dual-mode imaging [39,42,44,45,46,47,48]. Using bovine serum albumin (BSA) as templates, Liang and Xiao synthesized Gd-DTPA functionalized gold nanoclusters (BAG), which show intense red fluorescence emission (4.9 ± 0.8 quantum yield (%)) and high r_1_ (9.7 mM^−1^⋅s^−1^) [47]. The in vivo MRI demonstrates that BAG circulate freely in the blood pool with negligible accumulation in the liver and spleen and can be removed from the body through renal clearance, when BAG were injected intravenously into a Kunming mouse at a dose of 0.008 mmol Gd kg^−1^ via the tail vein. The unique properties of BAG make it an ideal dual-mode fluorescence/MR imaging contrast agent, suggesting its potential in practical bioimaging applications in the future. Very recently, Basilion and coauthors constructed a targeted nanocomposite, Au-Gd^3+^-prostate-specific membrane antigen (PSMA) NPs for MRI-guided radiotherapy of prostate cancer by immobilization of the Gd^3+^-complex and prostate-specific membrane antigen (PSMA) targeting ligands on the monodispersing Au NPs (as shown in Figure 4a) [50]. Because the hydrodynamic diameter of Au-Gd^3+^-PSMA NPs is 7.8 nm, Au-Gd^3+^-PSMA NPs enable efficient accumulation into the tumor site through the EPR effect and ligand-antigen binding, and can be excreted through the renal route. The r_1_ of Au-Gd^3+^-PSMA NPs (20.6 mM^−1^⋅s^−1^) is much higher than that of free Gd^3+^-complexes (5.5 mM^−1^⋅s^−1^). In addition, both of the Au and Gd^3+^ atoms can serve as sensitizers of radiotherapy. The Au-Gd^3+^-PSMA NPs show good tumor-targeting specificity, high MR contrast, significant in vivo radiation dose amplification, and renal clearance ability, which exhibit great potential in the clinical MR-guided radiotherapy of PSMA-positive solid tumors ((as shown in Figure 4b).

As a rising star of carbon nanomaterial, carbon quantum dots (CDs) have drawn tremendous attention because of their excellent optical property, high physicochemical stability, good biocompatibility, and the ease of surface functionalization [53,54,55,56,57]. Recently, various Gd^3+^-doped CDs (Gd-CD) have been synthesized for fluorescence/MR dual-mode imaging by the low temperature pyrolysis of precursors containing Gd^3+^ (such as Gd^3+^-complexes) and carbon [58,59,60,61]. Zou and coauthors reported Gd-CD-based theranostics for MRI-guided radiotherapy of a tumor (as shown in Figure 5) [58]. The Gd-CDs were synthesized through a one-pot pyrolysis of glycine and Gd-DTPA at 180 °C, which exhibited stable photoluminescence (PL) at the visible region, relatively long circulation time (~6 h), and efficient passive tumor-targeting ability. The r_1_ value of Gd-CDs was calculated to be 6.45 mM^−1^⋅s^−1^, which was higher than that of Magnevist^T^ (4.05 mM^−1^⋅s^−1^) under the same conditions. An in vivo experiment demonstrated that the Gd-CDs could provide better anatomical and pathophysiologic detection of tumor and precisely positioning for MRI-guided radiotherapy, when the mice were injected intravenously with the Gd-CD solution at a dose of 10 mg Gd kg^−1^. In addition, the efficient renal clearance of Gd-CDs meant it was finally excreted from the body by urine.

## 3. Paramagnetic Metal Nanodots

During the last two decades, several MNDs have been synthesized and used as MRCAs because of their advanced imaging properties compared to small molecule Gd^3+^-complexes. For example, the MNDs exhibit high surface-to-volume ratios, which allow the ^1^H to interact with a large number of paramagnetic ions in a tiny volume, resulting in high signal-to-noise ratios at ROI. It has been demonstrated that the size of MND is directly associated with the MRI contrast capability, biodistribution, blood circulation time, and clearance rate [5,14,15,18,23,25,27]. In order to finely control over the aforementioned parameters, the as-developed MND-based MRCAs for clinical applications are highly required to be monodisperse. The hydrophilic MNDs can be directly synthesized through reactions of paramagnetic metal ionic precursors with hydrophilic coating/functionalizing agents. Although this approach is simple and direct, it is difficult to control size and uniformity of MNDs in the aqueous synthetic methods. In order to achieve narrow size distribution and low crystalline defect, most of the MNDs are synthesized by nonaqueous synthesis routes. Because of their inherent hydrophobicity, MNDs should be coated/functionalized with hydrophilic and biocompatible ligands (shells) for biomedical applications. The post-synthetic modification strategies generally involve ligand exchange with hydrophilic molecules as well as encapsulation by hydrophilic shells.

### 3.1. Gadolinium Nanodots

Generally, there are two protocols for the synthesis of inorganic Gd^3+^ NDs: (1) direct synthesis of hydrophilic Gd^3+^ NDs in water or polyol using a stabilizing agent which allows for crystal growth, followed by ligand exchange with a more robust stabilizing agent to improve the colloidal stability of Gd^3+^ NDs in complex matrixes [62,63,64,65,66,67,68,69,70,71,72,73,74,75,76,77,78,79,80,81], and (2) preparation of hydrophobic Gd^3+^ NDs in high boiling organic solvents by the pyrolysis methods, and subsequent transfer of the hydrophobic Gd^3+^ NDs into aqueous phase by using a hydrophilic/amphiphilic ligand to render them water-dispersible [82,83,84,85,86,87,88,89,90].

#### 3.1.1. Gadolinium Oxide Nanodots

Several methods have been developed to synthesize Gd_2_O_3_ NDs for the application in MRCAs [62,63,64,65,66,67,68,69,70,71,72,74,77,78,79,80]. As early as 2006, Uvdal and coauthors synthesized Gd_2_O_3_ NDs (5 to 10 nm in diameter) by thermal decomposition of Gd^3+^ precursors (Gd(NO_3_)_3_·6H_2_O or GdCl_3_·6H_2_O) in the diethylene glycol (DEG) [62]. Both r_1_ and r_2_ of the DEG capped Gd_2_O_3_ NDs are approximately 2 times higher than those of Gd-DTPA. Lee’s group developed a one-pot synthesis method with two steps for preparing a series of hydrophilic Gd_2_O_3_ NDs with high colloidal stability: (1) using GdCl_3_·6H_2_O as Gd^3+^ as precursors, Gd_2_O_3_ NDs were firstly synthesized in tripropylene/triethylene glycol (TPG/TEG) under alkaline condition, and aged by H_2_O_2_ and/or an O_2_ flow; (2) the as-prepared Gd_2_O_3_ NDs were then stabilized by different coating materials including amino acid, polymers, and carbon [63,64,65,66,67,68,69,70]. As early as 2009, Lee and coauthors synthesized d-glucuronic acid coated Gd_2_O_3_ NDs in TPG by using d-glucuronic acid as stabilizer [63]. The d-glucuronic acid coated Gd_2_O_3_ NDs with 1.0 nm in diameter have a high r_1_ (9.9 mM^−1^⋅s^−1^), which can easily cross the brain-blood barrier (BBB), causing a strong contrast enhancement in the brain tumor within two hours. Due to the excretion of the injected d-glucuronic acid coated Gd_2_O_3_ NDs, the MRI contrast began to decrease at 2 h post-injection. The in vivo MR images of brain tumor, kidney, bladder, and aorta demonstrate that d-glucuronic acid coated Gd_2_O_3_ NDs have good tumor-targeting ability, strong blood pool effect, and high renal clearance when the mice were injected intravenously Gd_2_O_3_ NDs solution at a dose of 0.07 mmol Gd kg^−1^. Subsequently, Lee and coauthors successfully synthesized hydrophilic polyacrylic acid (PAA) coated Gd_2_O_3_ NDs (average diameter = 2.0 nm) with high biocompatibility by electrostatically binding of -COO^−^ groups of PAA with Gd^3+^ on the Gd_2_O_3_ NDs (as shown in Figure 6) [69]. Because the magnetic dipole interaction between surface Gd^3+^ and ^1^H is impeded by the PAA, both r_1_ and r_2_ values of Gd_2_O_3_ NDs are decreased with increasing the molecular weight of PAA (Mw = 1200, 5100 and 15,000 Da). Very recently, Lee and coauthors synthesized a kind of carbon-coated Gd_2_O_3_ NDs (Gd_2_O_3_@C, average diameter = 3.1 nm) by using dextrose as a carbon source in the aqueous solution (as shown in Figure 7) [70]. The Gd_2_O_3_@C have excellent colloidal stability, very high r_1_ value (16.26 mM^−1^⋅s^−1^, r_2_/r_1_ = 1.48), and strong photoluminescence (PL) in the visible region, which can be used as renal clearable MR/PL dual-mode imaging agent.

#### 3.1.2. NaGdF_4_ Nanodots

Hydrophobic NaGdF_4_ NDs with narrow size distribution are readily synthesized by pyrolysis methods via a high boiling binary solvent mixture [82,83,84,85,86,87,88,89]. In the presence of oleic acid, the approach has been demonstrated to be able to produce NaGdF_4_ NDs with high quality (e.g., low crystal defect and good monodispersity) through the thermal decomposition of Gd^3+^ precursors, NaOH, and NH_4_F in octadecene. In order to transfer the as-prepared hydrophobic NaGdF_4_ NDs into aqueous phase, the NaGdF_4_ NDs surfaces should be capped with appropriate surface-coating materials such as amphiphilic polymers and biomacromolecules. As early as 2011, Veggel and coauthors reported a size-selective synthesis of paramagnetic NaGdF_4_ NPs with four different sizes (between 2.5 and 8.0 nm in diameter) with good monodispersity by adjusting the concentration of the coordinating ligand (i.e., oleic acid), reaction time, and temperature [82]. The oleate coated NaGdF_4_ NPs were then transferred into aqueous phase by using polyvinylpyrrolidone (PVP) as phase transfer agent. They found that the r_1_ values of PVP coated NaGdF_4_ NPs are decreased from 7.2 to 3.0 mM^−1^⋅s^−1^ with increasing the NP size from 2.5 to 8.0 nm. In particular, the r_1_ value of PVP coated 2.5 nm NaGdF_4_ NDs is about twice as high as that of Magnevist^T^ under same conditions.

Shi and coauthors successfully employed the amphiphilic molecule, PEG-phospholipids, (such as DSPE-PEG_2000_ (1,2-Distearoyl-sn-glycero-3-phosphoethanolamine-Poly(ethylene glycol)_2000_) and/or the mixture of DSPE-PEG_2000_ and DSPE-PEG_2000_-NH_2_ (amine functionalized DSPE-PEG_2000_)) to convert 2 nm NaGdF_4_ NDs from hydrophobic to hydrophilic through the van der Waals interactions of the two hydrophobic tails of phospholipid groups of PEG-phospholipids and the oleic acids on the ND surface [84,85]. The PEG-phospholipids/amine functionalized PEG-phospholipids coated NaGdF_4_ NDs can be further functionalized by other molecules via suitable physical and chemical reactions. For example, poly-l-lysine (PLL) coated NaGdF_4_ NDs (NaGdF_4_@PLL NDs) were developed as dual-mode MRCA by layer-by-layer (LbL) self-assembly of positively charged PLL on the negatively charged DSPE-PEG_5000_ coated NaGdF_4_ NDs (as shown in Figure 8) [85]. The NaGdF_4_@PLL ND exhibits high r_1_ (6.42 mM^−1^⋅s^−1^) for T_1_-weighted MRI, while the PLL shows an excellent sensitive chemical exchange saturation transfer (CEST) effect for pH mapping (at +3.7 ppm). The results of in vivo small animal experiments demonstrate that NaGdF_4_@PLL NDs can be used as highly efficient MRCA for precise measurement of in vivo pH value and diagnosis of kidney and brain tumor. Moreover, the NaGdF_4_@PLL NDs could be excreted through urine with negligible toxicity to body tissues, which holds great promise for future clinical applications.

Taking benefit from robust Gd^3+^-phosphate coordination bonds, our group developed a facile method for transferring hydrophobic NaGdF_4_ NDs into aqueous phase though ligand exchange reaction between oleate and phosphopeptides in tryptone [86,87]. The tryptone-coated NaGdF_4_ NDs (tryptone-NaGdF_4_ NDs) have excellent colloidal stability, low toxicity, outstanding MRI enhancing performance (r_1_ = 6.745 mM^−1^⋅s^−1^), efficient renal clearance, and EPR effect-based passive tumor-targeting ability. Importantly, the tryptone-NaGdF_4_ NDs can be easily functionalized by other molecules including organic dyes and bioaffinity ligands. As shown in Figure 9, the tryptone-NaGdF_4_ NDs were further functionalized by hyaluronic acid (HA, a naturally occurring glycosaminoglycan) [87]. The HA functionalized tryptone-NaGdF_4_ NDs (tryptone-NaGdF_4_ ND@HAs) exhibit high binding affinity with CD44-positive cancer cells, good paramagnetic property (r_1_ = 7.57 mM^−1^⋅s^−1^) and reasonable biocompatibility. Using MDA-MB-231 tumor-bearing mouse as a model, the in vivo experimental results demonstrate that the tryptone-NaGdF_4_ ND@HAs cannot only efficiently accumulate in tumor (ca. 5.3% injection dosage (ID) g^−1^ at 2 h post-injection), but also have an excellent renal clearance efficiency (ca. 75% ID at 24 h post-injection). The approach provides a useful strategy for the preparation of renal clearable MRCAs with positive tumor-targeting ability. Using a similar phase transferring principle, the peptide functionalized NaGdF_4_ NDs (pPeptide-NaGdF_4_ NDs) were prepared by conjugation of the hydrophobic oleate coated NaGdF_4_ NDs (4.2 nm in diameter) with the mixture of phosphorylated peptides including a tumor targeting phosphopeptide (pD-SP5) and a cell penetrating phosphopeptide (pCLIP6) [88]. Due to its high isoelectric point, the pCLIP6 can enhanced the cellular uptake of pPeptide-NaGdF_4_ NDs. The pD-SP5 can improve the tumor-targeting ability of pPeptide-NaGdF_4_ NDs because it has high binding affinity to human tumor cells. The pPeptide-NaGdF_4_ NDs show low toxicity, outstanding MRI enhancing performance (r_1_ = 13.2 mM^−1^⋅s^−1^) and positive-tumor targeting ability, which were used successfully as efficient MRCA for an in vivo imaging small drug induced orthotopic colorectal tumor (c.a., 195 mm^3^), when the mice were then intravenously injected with pPeptide-NaGdF_4_ NDs at a dose of 5 mg Gd kg^−1^. In addition, the half-life of pPeptide-NaGdF_4_ NDs in blood is found to be 0.5 h, and more than 70% Gd is excreted with the urine after 24 h intravenous administration.

### 3.2. Iron Nanodots

In 1996, the U. S. FDA approved Ferumoxides (SPIONs) as T_2_-weighted MR CAs for the diagnosis of liver disease [91]. After that, various strategies based on physical, chemical, and biological methods have been developed for synthesizing the SPIONs including FeNDs for using as T_2_-weighted MRCAs [91,92,93,94,95]. Because of the negative contrast effect and magnetic susceptibility artifacts, it is still a great challenge when the SPION-enhanced T_2_-weighted MRI is employed to distinguish the lesion region in the tissues with low background MR signals such as bone and vasculature. Recently, several methods have developed for synthesis of FeNDs, which are showing increasing potential as alternatives to Gd^3+^-based T_1_-weighted MRCAs because of low magnetization by a strong size-related surface spin-canting effect [96,97,98,99,100,101,102,103,104,105,106,107,108,109,110,111]. For instance, Wu and coauthors synthesized a silica-coated Fe_3_O_4_ ND (4 nm in diameter), which exhibited a good r_1_ relaxivity of 1.2 mM^−1^⋅s^−1^ with a low r_2_/r_1_ ratio of 6.5 [101]. The result of in vivo T_1_-weighted MR imaging of heart, liver, kidney, and bladder in a mouse demonstrated that silica-coated FeNDs exhibited strong MR enhancement capability, when the mice were then intravenously injected with silica-coated FeNDs at a dose of 2.8 mg Fe kg^−1^. Shi and coauthors reported a zwitterion l-cysteine (Cys) coated Fe_3_O_4_ ND (3.2 nm in diameter, Fe_3_O_4_-PEG-Cys) with r_1_ relaxivity of 1.2 mM^−1^⋅s^−1^ [102]. The Fe_3_O_4_-PEG-Cys are able to resist macrophage cellular uptake, and display a prolonged blood circulation time with a half-life of 6.2 h. In vivo experimental results demonstrated that the Fe_3_O_4_-PEG-Cys were able to be used as a T_1_-weighted MRCA for enhanced blood pool and tumor MR imaging, when the mice were then intravenously injected with Fe_3_O_4_-PEG-Cys at a dose of 0.05 mmol Fe kg^−1^. Bawendi and coauthors prepared zwitterion-coated ultrasmall superparamagnetic Fe_2_O_3_ NDs (ZES-SPIONs) with hydrodynamic size of 5.5 nm for use as Gd^3+^ free T_1_-weighted MRCA [103]. The ZES-SPIONs have strong T_1_-weighted MRI enhancement capacity (r_1_ = 5.2 mM^−1^⋅s^−1^), and the majority of ZES-SPIONs are cleared through the renal route within 24 h intravenous administration at a dose of 0.2 mmol Fe kg^−1^. As shown in Figure 10, Zhang and coauthors reported a large-scalable (up to 10 L) albumin-constrained strategy to synthesize monodispersed 3 nm ferrous sulfide NDs (FeS@BSA) with an ultralow magnetization by reaction of the mixture of BSA and FeCl_2_ with Na_2_S under ambient conditions (pH 11 and 37 °C) [104]. In this case, BSA plays crucial roles in the synthesis process including as a constrained microenvironment reactor for particle growth, a water-soluble ligand for colloidal stability, and a carrier for multifunctionality. FeS@BSAs exhibit high MRI enhancing performance (r_1_ = 5.35 mM^−1^⋅s^−1^), good photothermal conversion efficiency (η = 30.04%), strong tumor-targeting ability, and an efficient renal clearance characteristic. In vivo experiments show that FeS@BSAs have good performance of T_1_-weighted MR/phototheranostics dual-mode imaging-guided photothermal therapy (PTT) of mouse-bearing 4T1 tumor, demonstrating FeS@BSA to be an efficient T_1_-weighted MR/PA/PTT theranostic agent.

Fe^3+^ coordination polymer NDs (Fe-CPNDs) were synthesized through the self-assembling the multidental Fe^3+^-polyvinyl pyrrolidone (PVP) complexes and Fe^3+^-gallic acid (GA) complexes and used as nanotheranostics for MRI-guided PTT [107,108,109]. For instance, our group has developed a simple and scalable method for synthesizing the pH-activated Fe coordination polymer NDs (Fe-CPNDs) by the coordination reactions among Fe^3+^, GA, and PVP at ambient conditions (as shown in Figure 11) [107]. The Fe-CPNDs exhibit an ultrasmall hydrodynamic diameter (5.3 nm), nearly neutral Zeta potential (−3.76 mV), pH-activatable MRI imaging contrast (r_1_ = 1.9 mM^−1^⋅s^−1^ at pH 5.0), and outstanding photothermal performance. The Fe-CPNDs were successfully used as T_1_-weighted MRCA to detect mouse-bearing tumors as small as 5 mm^3^ in volume, and as a PTT agent to completely suppress tumor growth by MRI-guided PTT, demonstrating that Fe-CPNDs constitute a new class of renal clearable theranostics. In addition, the Fe-CPND-enhanced MRI was successfully employed for noninvasively monitoring the kidney dysfunction by drug (daunomycin)-induced kidney injury, which further highlights the potential in clinical applications of the Fe-CPND [108].

Development of contrast agents for simultaneously T_1_-/T_2_-weighted dual-mode MRI may circumvent the drawbacks of single imaging modalities. Iron-based NDs are also able to serve as T_1_-/T_2_-weighted dual-mode MRCAs [110,111,112,113]. Uvdal and coauthors synthesized a series of water-dispersible PAA-coated Fe_3_O_4_ NDs via a modified one-step coprecipitation approach [110]. In particular, the 2.2 nm PAA coated Fe_3_O_4_ NDs have relatively high relaxivities (r_1_ = 6.15 mM^−1^⋅s^−1^ and r_2_ = 28.62 mM^−1^⋅s^−1^) and low r_2_/r_1_ ratio (4.65). Using a mouse model, the in vivo experiments indicate that the 2.2 nm PAA-coated Fe_3_O_4_ NDs exhibit long-term circulation, low toxicity, and great contrast enhancement (brightened on the T_1_-weighted and darkened on the T_2_-weighted MR images), when the mice were then intravenously injected with PAA-coated Fe_3_O_4_ NDs with a dose of 0.0125 mmol Fe kg^−1^. The in vivo results demonstrate that the 2.2 nm PAA-coated Fe_3_O_4_ NDs have great potential as T_1_-/T_2_-weighted dual-mode MRCA for clinical applications including diagnosis of renal failure, myocardial infarction, atherosclerotic plaque, and tumor. Very recently, Shi and coauthors developed a strategy for preparing switchable T_1_/T_2_-weighted dual-mode MRCA by formation of cystamine dihydrochloride (Cys) cross-linked Fe_3_O_4_ NDs clusters [113]. The Fe_3_O_4_ NDs clusters, with a hydrodynamic size of 134.4 nm, and can be dissociated to single 3.3 nm Fe_3_O_4_ NDs under a reducing microenvironment (e.g., 10 mmol L^−1^ glutathione (GSH)) because of redox-responsiveness of the disulfide bond of Cys. The Fe_3_O_4_ NDs clusters exhibit a dominant T_2_-weighted MR effect with an r_2_ of 26.4 mM^−1^⋅s^−1^, while Fe_3_O_4_ NDs have a strong T_1_-weighted MR effect with an r_1_ of 3.9 mM^−1^⋅s^−1^. Due to the reductive tumor microenvironment, the Fe_3_O_4_ NCs can be utilized for dynamic precision imaging of a subcutaneous tumor model in vivo, and pass through the kidney filter, when the mice were then intravenously injected Fe_3_O_4_ NCs with a dose of 2.5 mmol Fe kg^−1^.

### 3.3. Other Paramagnetic Metal-Based Nanomaterials

Due to their in vivo safety, Mn^2+^-based T_1_-weighted MRCAs have attracted increasing attention [114,115,116,117,118,119,120,121,122,123,124,125,126,127,128,129,130,131,132,133,134]. Although free Mn^2+^ has a higher r_1_ than those of Mn-based nanomaterials, Mn^2+^ exhibits low accumulation and poor performance for disease contrast because of its short blood retention time in vivo [114,115,116,117,118,119]. The increase of Mn^2+^ accumulation in a tumor can be achieved through the design of pH/GSH-activated Mn-based nanomaterials because a tumor has weakly acidic microenvironment and high concentration of GSH, and nanomaterials can efficiently accumulate in a tumor by EPR effect [123,124,125,126,127,128,129]. Therefore, several tumor microenvironment activatable Mn-based nanotheranostic systems have been constructed for T_1_-weighted MRI-guided therapy. We have synthesized the polydopamine@ultrathin manganese dioxide/methylene blue nanoflowers (PDA@ut-MnO_2_/MB NFs) for the T_1_-weighted MRI-guided PTT/photodynamic therapy (PDT) synergistic therapy of tumor (as shown in Figure 12) [125]. In the presence of 5 mmol⋅L^−1^ GSH, the r_1_ of PDA@ut-MnO_2_/MB NFs is significantly increased from 0.79 to 5.64 mM^−1^⋅s^−1^ since the ultrathin MnO_2_ nanosheets on PDA@ut-MnO_2_/MB NFs can be reduced into Mn^2+^ ions by GSH. Due to the performance in response to the tumor microenvironment, the PDA@ut-MnO_2_/MB NFs show an enhancement 3 times that of the T_1_-weighted MRI signal at the tumor site at 4 h post-injection. The Mn^2+^ can be diffused from the tumor to the circulatory system and excreted from the body through renal clearance. In addition, doping of Mn^2+^ into the matrix of other metallic NDs and/or formation of Mn^2+^-complexes/nanocomposites can also improve its T_1_-weighted MR contrast ability [115,130,131,132,133,134]. For example, Wang and coauthors developed a Mn^2+^-based MRCA (MNP-PEG-Mn) through chelation of Mn^2+^ with 5.6 nm water-soluble melanin NDs [134]. The r_1_ (20.56 mM^−1^⋅s^−1^) of as-prepared MNP-PEG-Mn is much higher than that of Gadodiamide (6.00 mM^−1^⋅s^−1^). Using a 3T3 tumor-bearing mouse as model, in vivo MRI experiments demonstrated that MNP-PEG-Mn (200 μL of 8 mg mL^−1^ MNP-PEG-Mn PBS solution) showed excellent tumor-targeting specificity, and could be efficiently excreted via renal and hepatobiliary pathways with negligible toxicity.

Because Dy^3+^ and Ho^3+^ ions exhibit relatively high magnetic moments among of lanthanide (III) (Ln^3+^) ions, Dy and Ho NDs are believed as promising candidates for T_2_-weighted MRCAs with renal excretion [135,136,137,138,139]. As early as 2011, Lee and coauthors developed a facile one-pot method for synthesis of d-glucuronic acid-coated Ln_2_O_3_ NDs (Ln = Eu, Gd, Dy, Ho, and Er) [138]. They demonstrated that the d-glucuronic acid-coated 3.2 nm Dy_2_O_3_ NDs have high r_2_ (65.04 mM^−1^⋅s^−1^) and very low r_1_ (0.008 mM^−1^⋅s^−1^). After injection of the Dy_2_O_3_ NDs at a dose of 0.05 mmol Dy kg^−1^ through tail vein of mouse, the clearly negative MR contrast enhancement in both liver and kidneys of mouse are observed in in vivo T_2_-weighted MRI. The Dy_2_O_3_ NDs are also excreted from the body through the renal route, which is prerequisite for clinical applications as a MRCA.

## 4. Dual Paramagnetic Metal Nanodots

The MR contrast capabilities of nanomaterials can be further improved while two paramagnetic metallic ions are integrated into one nanoplatform [140,141,142,143,144,145,146,147,148,149,150,151,152,153,154,155]. For instance, Zhou and coauthors constructed a 4.95 nm triple-mode imaging platform by doping of Mn^2+^ and ^68^gallium (III) (^68^Ga^3+^) into the matrix of copper sulfide (CuS) NDs using BSA as the synthetic template [143]. Although the Mn^2+^/^68^Ga^3+^-CuS@BSA NDs have relatively low r_1_ (0.1119 mM^−1^⋅s^−1^, the ratio of r_2_/r_1_ = 1.67), in vivo experimental results of SKOV-3 ovarian tumor-bearing mouse demonstrated that the as-prepared Mn^2+^/^68^Ga^3+^-CuS@BSA NDs (150 μL solution at 2 OD concentration of NDs) could be used as an excellent agent for T_1_-weihted MR/positron emission tomography (PET)/photoacoustic (PAT) triple-mode imaging-guided PTT of the tumor, and were efficiently cleared via the renal-urinary route. Very recently, our group constructed a multifunctional nanotheranostic (MnIOMCP) for active tumor-targeting T_1_-/T_2_-weighted dual-mode MRI-guided biological-photothermal therapy (bio-PTT) through bioconjugation of the monocyclic peptides (MCP, the CXC chemokine receptor 4 (CXCR4) antagonist) with 3.8 nm manganese-doped iron oxide NDs (MnIO NDs) [146]. The MnIOMCP displays reasonable T_1_-/T_2_-weighted MR contrast abilities (r_1_ = 13.1 mM^−1^⋅s^−1^, r_2_ = 46.6 mM^−1^⋅s^−1^ and r_2_/r_1_ = 3.56), good photothermal conversion efficiency (η = 28.8%), strong tumor-targeting ability (∼15.9% ID g^−1^ at 1 h after intravenous injection with a dose of 10 mg [Mn + Fe] kg^−1^) and clear inhibition of CXCR4-positive tumor growth. In addition, the MnIOMCP at can be rapidly excreted from the body through renal clearance (about 75% ID of MnIOMCP found in urine at 24 h post-injection), which illuminates a new pathway for the development of efficient nanotheranostics with high biosafety. In addition, the r_1_ and/or r_2_ values of Ln^3+^-based nanomaterials could be increased by mixing paramagnetic transition metal ions into them owing to unpaired 3d-electrons of transition metal ions [151,152,153,154,155]. Gao and coauthors reported a facile strategy to design and synthesis of zwitterionic dopamine sulfonate-coated 4.8 nm gadolinium-embedded iron oxide NDs (GdIO@ZDS), which showed a hydrodynamic diameter of about 5.2 nm in both PBS buffer and BSA solution [155]. The combination of the spincanting effects and the collection of Gd^3+^ within small-sized GdIO NDs led to a strongly enhanced T_1_-weighted MR contrast effect, which exhibited a high r_1_ of 7.85 mM^−1^⋅s^−1^ and a low r_2_/r_1_ ratio of 5.24. Using SKOV3-bearing mouse as a model, the in vivo experimental results demonstrated that the GdIO@ZDS with a dose of 2.0 mg GdIO@ZDS kg^−1^ are suitable candidates as excellent T_1_-weighted MRCAs for tumor imaging and disease diagnosis because they have relatively long circulation half-life (∼50 min), passive-tumor targeting capacity, and efficient renal clearance ability.

## 5. Conclusions and Outlook

In summary, we have illustrated the recent advances in the development of renal clearable MRCAs including Gd^3+^-based composites and MNDs. Compared to conventional small molecule Gd^3+^-complex contrast agents, the composites/MNDs have demonstrated improved MR signal intensity, targeting ability, and longer circulation time both in vitro and in small animal disease models, especially for cancer diagnosis. In particular, with the help of nanotechnology, the recent research developments have progressed towards construction of renal clearable MRCAs with multifunctionality, which enables us to integrate several functions at the same time, such as simultaneous disease targeting, multimodal imaging, and therapy. For instance, several unique characteristics of MNDs enable their use for selective cancer theragnostics, which include: (a) their size, which leads to preferential accumulation of MNDs in tumors though EPR effect, wide MRI time window by prolonging circulation time, and completely excreted from the body within a reasonable period of time (i.e., within a few of days) through renal clearance; (b) high surface-to-volume ratios, which result in strong enhancement of MR signal at ROI through increasing the interaction opportunities of ^1^H with paramagnetic ions; and (c) large surface area, which exhibits the possibility to load different molecular therapeutics for MRI-guided therapy and/or functionalize with cancer-homing ligands for achieving positive-tumor targeting. The performance of MND-based MRCAs could be further improved through optimizing one or more of the above-mentioned characteristics of the MNDs. The renal clearable composite-/MND-based MRCAs have, indeed, bright prospects regarding their possibilities for biomedical applications, which have already been demonstrated in the scientific literature.

Unfortunately, very few MND-based MRCAs were approved for clinical application except SPIONs (Ferumoxides). On the other hand, the need for specific molecular information is greater than ever, since we are entering into the era of precision medicine. The noninvasive diagnostic or screening methods such as MRI can efficiently help patients to avoid ineffective and/or costly treatments because many of the newly developed powerful and expensive therapies are only effective in a subset of patients. This situation strongly requires the development of high-performance contrast agents to improve the accuracy of molecular imaging. The clinical translation of MND-based MRCAs research is generally impeded by the significant heterogeneity in the construction of these agents. Up to date, the MND-based MRCAs are still in the experimental stage. There are several technical challenges regarding the MND-based MRCAs for clinical trials that need to be clearly addressed before they are evaluated in humans. For example, the safety and efficacy of the MND-based MRCAs should be comprehensively evaluated. Ongoing research should focus on evaluating the biodistribution, pharmacokinetics, and ultimate fates in vivo, improving targeting specificity while minimizing toxicity, and demonstrating the translational potential with appropriate animal models. It is critical to develop cost-effective methods for the kilogram-scale production of MND-based MRCAs since the MRCAs are normally administered in gram quantities. This matter could be solved by systematic optimization of synthesis conditions of polyol methods and thermal decomposition methods. The leakage of Gd^3+^ should be minimized while Gd^3+^NDs were used as MRCAs. Future efforts should aim to synthesize chelating agents with a high Gd^3+^ binding constant, and/or develop coating materials with highly stable physicochemical properties and excellent biocompatibility. In addition, integration of two or more paramagnetic metallic elements into a single hybrid ND is beneficial to circumvent their individual drawbacks and potentially provide more comprehensive imaging information through T_1_-/T_2_-weighted dual-mode MRI. The MNDs are providing revolutionary potential as new MRCAs, which could achieve various clinical applications through close cooperation among of multidisciplinary teams of chemists, materials scientists, biologists, pharmacists, physicians, and imaging experts, and have a strongly positive impact on human health.

## Figures and Tables

**Figure 1 molecules-25-05072-f001:**
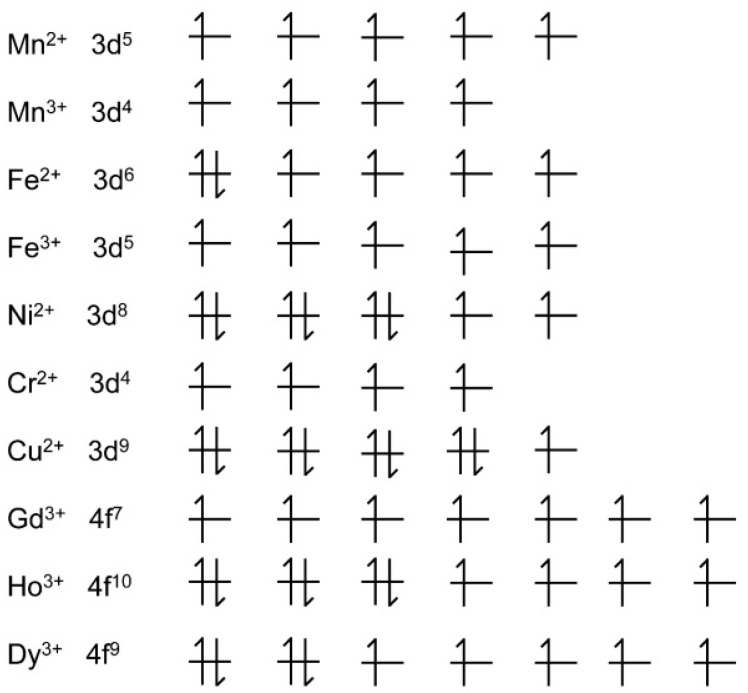
The electron subshell diagrams of typical paramagnetic cations. Generally, the larger number of unpaired electrons leads to stranger magnetic resonance (MR) contrast.

**Figure 2 molecules-25-05072-f002:**
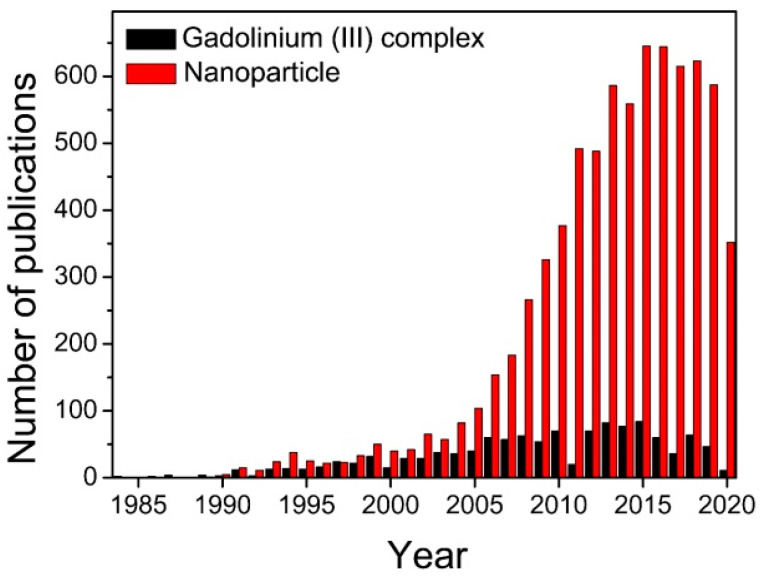
The number of publications searching for “magnetic resonance imaging and contrast agents” plus “gadolinium (III) complex” or “nanoparticle” in the “Web of Science”. Development of nanoparticle-based MRCAs has been the hot topic of the area since 2005.

**Figure 3 molecules-25-05072-f003:**
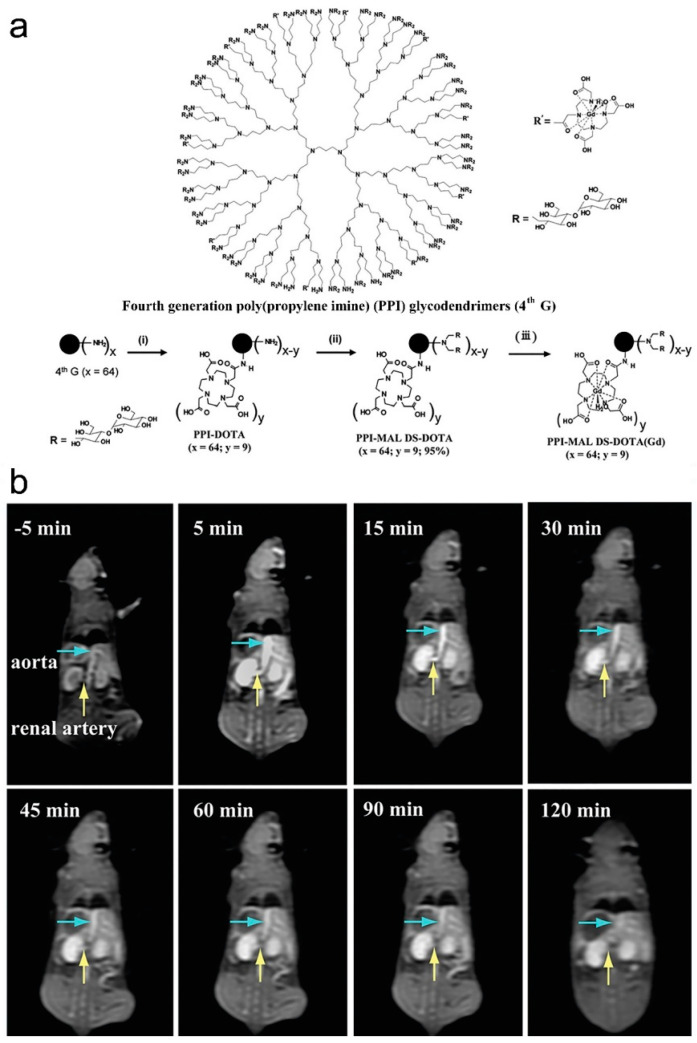
(**a**) Schematic illustration of the approach used to synthesize the PPI-MAL DS-DOTA(Gd), and (**b**) T_1_-weighted MR images of mouse aorta and renal artery at 5 min before injection (−5 min) and at 15, 30, 45, 60, 90, and 120 min post-injection of the PPI-MAL DS-DOTA(Gd) ([Gd^3+^] = 2 mg mL^−1^ in 0.2 mL saline through tail vein, adapted from Shi et al. 2016 [36], Copyright 2016 The Royal Society of Chemistry and reproduced with permission).

**Figure 4 molecules-25-05072-f004:**
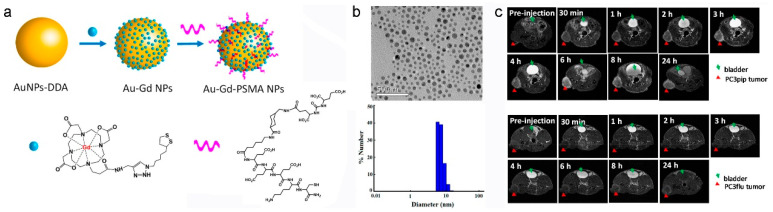
Au-Gd^3+^-prostate-specific membrane antigen (PSMA) NPs for MR-guided radiation therapy. (**a**) Schematic representation of Au-Gd^3+^-PSMA NPs. (**b**) TEM micrograph indicates that the average core size of Au-Gd^3+^-PSMA NPs is 5 nm, and DLS shows that the hydrodynamic diameter of Au-Gd^3+^-PSMA NPs is 7.8 nm. (**c**) In vivo tumor targeting of Au-Gd^3+^-PSMA NPs and MR imaging of PC3pip tumor-bearing mouse (up) and PC3flu tumor-bearing mouse (bottom) obtained at 7 T. PC3pip tumor cell expresses high level of PSMA, while PC3flu tumor cell expresses low level of PSMA (the mice were injected with Au-Gd^3+^-PSMA NPs at 60 μmol Gd^3+^⋅kg^−1^ through the tail vein, adapted from Basilion et al. 2020 [50], Copyright 2020 The American Chemical Society and reproduced with permission).

**Figure 5 molecules-25-05072-f005:**
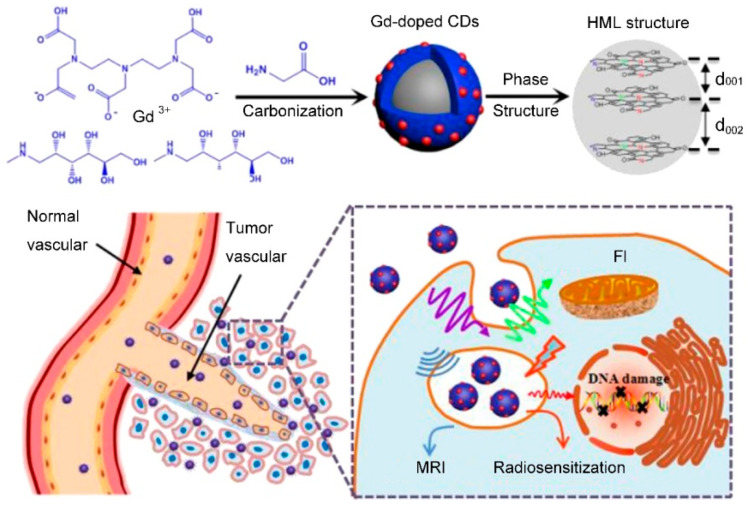
Schematic synthesis and application of Gd-doped CDs through one-pot pyrolysis of glycine and Gd-DTPA at 180 °C (adapted from Zou et al. 2017 [58], Copyright 2017 The Elsevier Ltd. and reproduced with permission).

**Figure 6 molecules-25-05072-f006:**
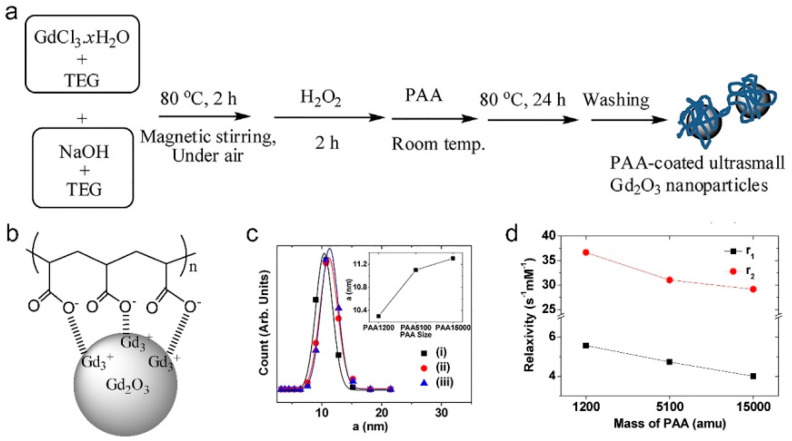
(**a**) Schematic representation of the one-pot synthesis of polyacrylic acid (PAA) coated Gd_2_O_3_ NDs, (**b**) the representative PAA surface-coating structure, (**c**) DLS measurement of (i) PAA 1200, (ii) PAA 5100, and (iii) PAA 15,000 coated Gd_2_O_3_ NDs, inset of (**c**) is the hydrodynamic diameter of PAA coated Gd_2_O_3_ NDs as a function of PAA molecular weight, and (**d**) r_1_ and r_2_ values as a function of PAA molecular weight (adapted from Lee et al. 2019 [69], Copyright 2017 the Elsevier B.V. and reproduced with permission).

**Figure 7 molecules-25-05072-f007:**
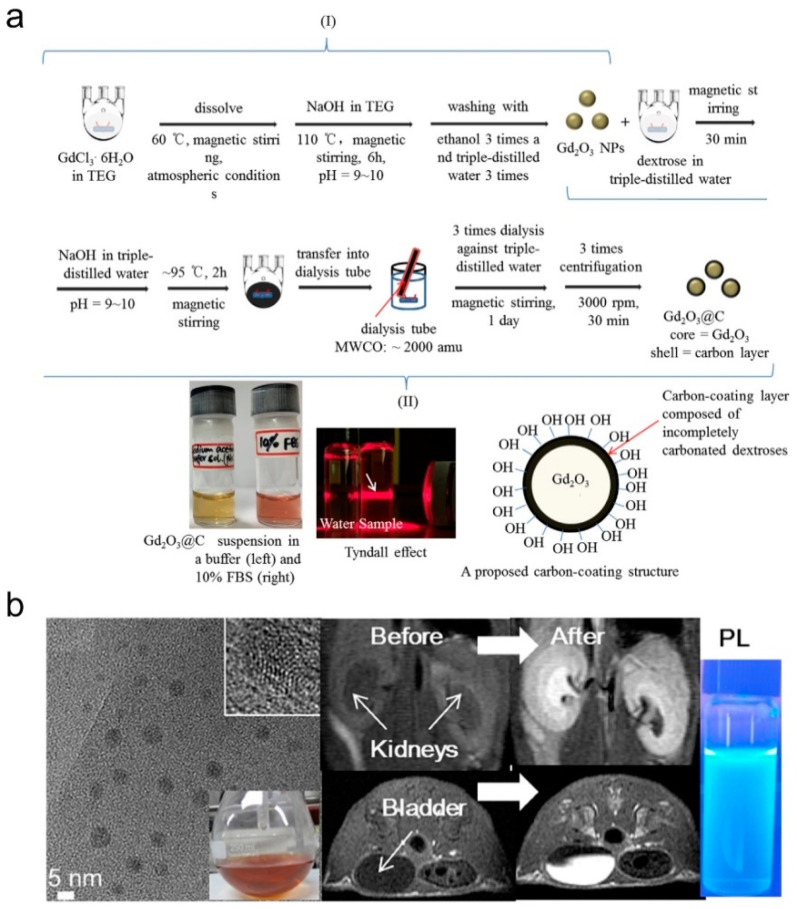
(**a**) Two-step synthesis of Gd_2_O_3_@C: (i) the synthesis of Gd_2_O_3_ NDs in triethylene glycol (TEG), and (ii) carbon coating on the Gd_2_O_3_ ND surfaces in aqueous solution. From left to right at the bottom of (**a**), photographs of the Gd_2_O_3_@C in sodium acetate buffer solution (pH = 7.0) and a 10% FBS in RPMI1640 medium, the Tyndall effect (or laser light scattering) as indicated with an arrow in the Gd_2_O_3_@C in water sample (right) with no light scattering in the reference triple-distilled water (left), and a proposed carbon-coating structure of Gd_2_O_3_@C. (**b**) High resolution transmission electron microscope (HRTEM) micrograph of Gd_2_O_3_@C, photograph of Gd_2_O_3_@C in water, in vivo MRI of mouse before or after intravenously administrated by Gd_2_O_3_@C and PL of Gd_2_O_3_@C at a dose of 0.1 mmol Gd kg^−1^ (adapted from Lee et al. 2020 [70], Copyright 2019 the Elsevier B.V. and reproduced with permission).

**Figure 8 molecules-25-05072-f008:**
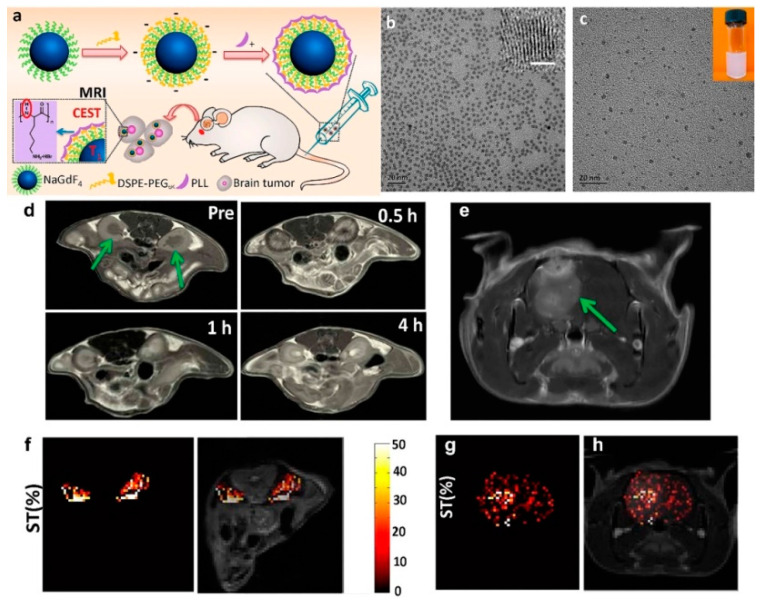
(**a**) Schematic representation of the synthesis of NaGdF_4_@PLL NDs for T_1_-weighted and chemical exchange saturation transfer (CEST) MR imaging. (**b**) TEM and HRTEM (inset, scale bar: 1 nm) micrograph oleic acid-coated NaGdF_4_ NDs in chloroform. (**c**) TEM micrograph and photograph (inset) of NaGdF_4_@PLL NDs in water. (**d**) In vivo T_1_-weighted MRI of kidneys of mouse (as arrowed) before and after the intravenous administration of NaGdF_4_@PLL NDs at a dose of 5 mg Gd kg^−1^. (**e**) In vivo T_1_-weighted MRI of brain tumor (as arrowed) after the intravenous administration of NaGdF_4_@PLL NDs at a dose of 10 mg Gd kg^−1^. (**f**) CEST contrast difference map between pre/post-injection following radio frequency (RF) irradiation at 3.0 μT. Only the kidney signal is displayed in color on the grayscale image to highlight the CEST effect. (**g**) CEST ST difference map between pre/post-injection at 3.0 μT. Only the brain ventricle signal is displayed in color on the grayscale image to highlight the CEST effect. (**h**) Merged image of (**e**) and (**g**) (adapted from Shi et al. 2016 [85], Copyright 2016 the American Chemical Society and reproduced with permission).

**Figure 9 molecules-25-05072-f009:**
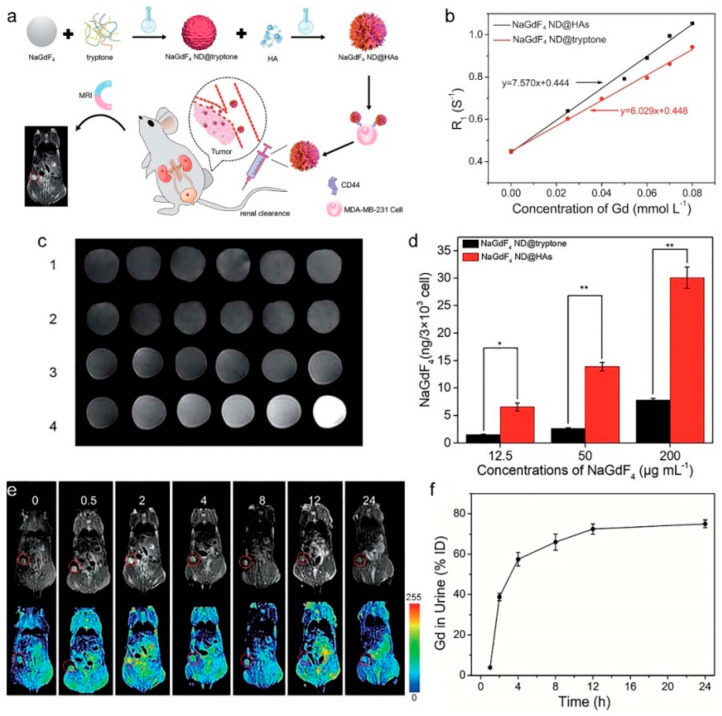
(**a**) Schematic representation of NaGdF_4_ ND@HAs synthesis, and the application in MRI of tumor through recognizing the overexpressed CD44 on cancer cell membrane. (**b**) R_1_ relaxivities of NaGdF_4_ ND@tryptone and NaGdF_4_ ND@HAs as a function of the molar concentration of Gd^3+^ in solution, respectively. (**c**) MR images of (1) NaGdF_4_ ND@HA-stained MCF-7 cells, (2) free HA + MDA-MB-231 cells + NaGdF_4_ ND@HAs, (3) NaGdF_4_ ND@tryptone-stained MDA-MB-231 cells, (4) NaGdF_4_ ND@HA-stained MDAMB-231 cells. (**d**) The amounts of Gd element in the NaGdF_4_ ND-stained MDA-MB-231cells. Error bars mean standard deviations (*n* = 5, * *p* < 0.05 or ** *p* < 0.01 from an analysis of variance with Tukey’s post-test). (**e**) In vivo MR images and corresponding pseudo color images of Balb/c mouse bearing MDA-MB-231 tumor after intravenous injection of NaGdF_4_ ND@HAs (10 mg Gd kg^−1^) at different timed intervals (0 (pre-injection), 0.5, 2, 4, 8, 12 and 24 h post-injection), respectively. (**f**) The total amounts of NaGdF_4_ ND@HAs in mouse urine as a function of post-injection times. Error bars mean standard deviations (*n* = 5) (adapted from Yan et al. 2020 [87], Copyright 2020 The Royal Society of Chemistry and reproduced with permission).

**Figure 10 molecules-25-05072-f010:**
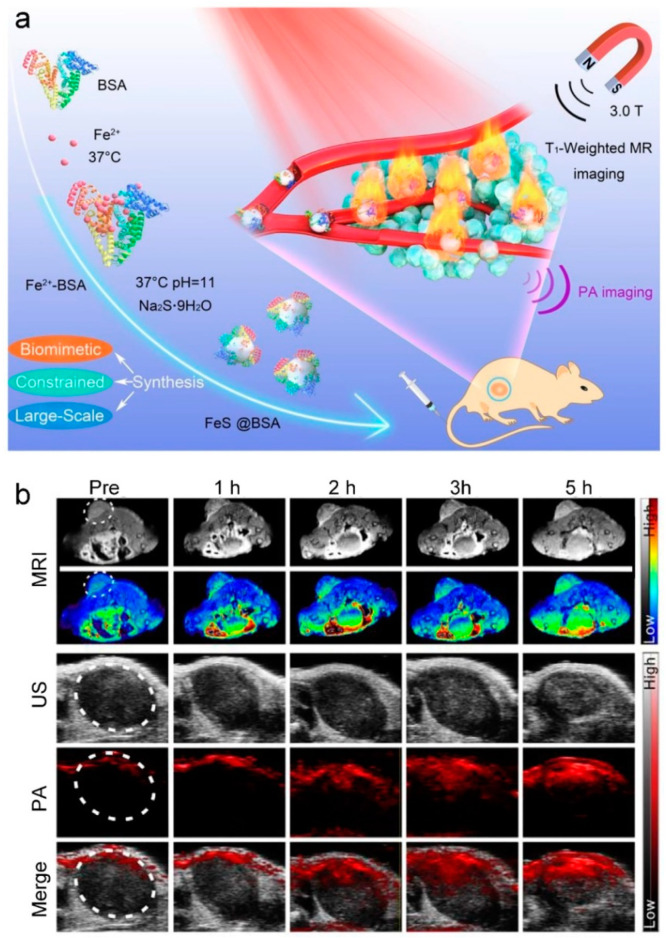
(**a**) Schematic representation of bovine serum albumin (BSA)-constrained biomimetic synthesis of 3 nm FeS@BSA for in vivo T_1_-weighted MR/phototheranostics dual-mode imaging-guided photothermal therapy (PTT) of tumors. (**b**) In vivo T_1_-weighted MR, US, PA, and merged (US and PA) images of tumor after intravenous injection of at a dose of 20 mg FeS@BSA kg^−1^ (adapted from Zhang et al. 2020 [104], Copyright 2020 Elsevier Ltd. and reproduced with permission).

**Figure 11 molecules-25-05072-f011:**
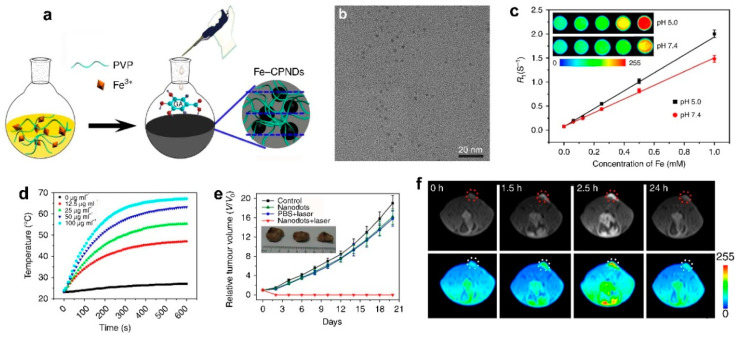
(**a**) Schematic representation of the synthesis of Fe-CPNDs. (**b**) TEM micrograph of Fe-CPNDs. (**c**) The effect of solution pH on the R_1_ relaxivity of Fe-CPNDs. (**d**) Temperature elevation of Fe-CPNDs solutions with various concentrations under 1.3 W cm^−2^ 808 nm NIR laser irradiation for 10 min. (**e**) Tumor growth curves of different groups of mice after intravenous treatments. The inset shows the digital photographs of tumors collected from different groups of mice at day 20th post-administration. (**f**) In vivo MR images of the SW620 tumor-bearing nude mouse after intravenous injection of at a dose of 0.25 mg Fe kg^−1^ (The tumor was marked by circle, which was about 5 mm^3^ in volume.) after the intravenous injection of Fe-CPNDs at different time intervals (0 h indicates pre-injection) (adapted from Liu et al. 2015 [107], Copyright 2015 Macmillan Publishers Ltd. and reproduced with permission).

**Figure 12 molecules-25-05072-f012:**
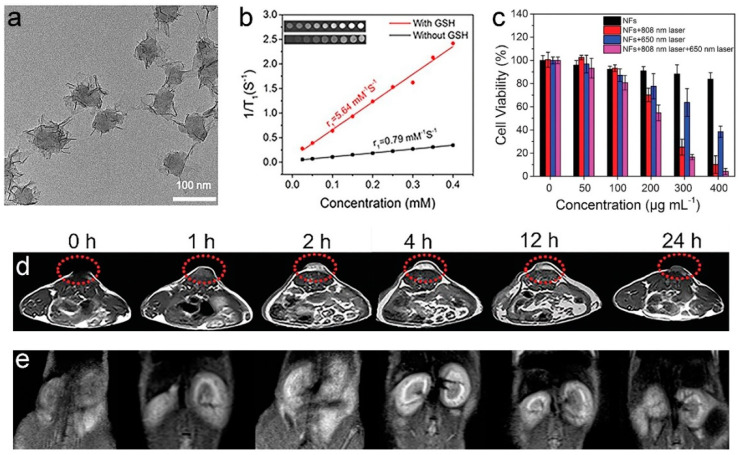
(**a**) TEM micrographs of PDA@ut-MnO_2_/MB NFs. (**b**) The r_1_ values of PDA@ut-MnO_2_/MB NFs with and without 5 mmol L^−1^ GSH. (**c**) An assessment of PDT/PTT efficacy using PDA@ut-MnO_2_/MB NFs via CCK-8 assays. In vivo MR images of HCT 116 tumor-bearing mouse (**d**) tumor and (**e**) kidneys different time points after the intravenous injection of PDA@ut-MnO_2_/MB NFs (0, 1, 2, 4, 12, and 24 h; 0 h means pre-injection) at a dose of 10 mg Mn kg^−1^ (adapted from Sun et al. 2019 [125], Copyright 2019 The Royal Society of Chemistry and reproduced with permission).

**Table 1 molecules-25-05072-t001:** Typical magnetic resonance imaging contrast agents (MRCAs) clinically approved or in clinical trials.

Trade Name	Generic Name	Chemical Code	MRI Mode	Clinical Trial	Clinically Approved
Dotarem/Clariscan	Gadoterate meglumine	Gd-DOTA	T_1_-weighted	-	Yes
ProHance	Gadoteridol	Gd-HPDO3A	T_1_-weighted	-	Yes
Gadovist	Gadobutrol	Gd-DO3A-butrol	T_1_-weighted	-	Yes
Magnevist	Gadopentetate dimeglumine	Gd-DTPA	T_1_-weighted	-	Yes
Omniscan	Gadodiamide	Gd-DTPA-BMA	T_1_-weighted	-	Yes
Optimark	Gadoversetamide	Gd-DTPA-BMEA	T_1_-weighted	-	Yes
Multihance	Gadobenate dimeglumine	Gd-BOPTA	T_1_-weighted	-	Yes
Combidex/Sinerem	Ferumoxtran	Dextran coated SPION	T_2_-weighted	Yes	-
Resovist/Cliavist	Ferucarbotran/Ferrixan	Carboxydextran coated SPION	T_2_-weighted	-	Yes
Feridex I.V./Endorem	Ferumoxide	Dextran	T_2_-weighted	-	Yes
Feraheme/Rienso	Ferumoxytol	Carboxymethyl-dextran coated SPION	T_2_-weighted	-	Yes
Clariscan	Feruglose	PEGylated starch coated SPION	T_2_-weighted	Yes	-
Lumirem/GastroMARK	Ferumoxsil	Siloxane coated SPION	T_2_-weighted	-	Yes
Abdoscan	-	Sulfonated poly (styrene-divinylbenzene) copolymer coated SPION	T_2_-weighted	-	Yes

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
