# Peer review of "The Renal Clearable Magnetic Resonance Imaging Contrast Agents: State of the Art and Recent Advances"

_molecules, 2020, doi:10.3390/molecules25215072_

Round 1

Reviewer 1 Report

In general, this paper represents a good overview of the current state of the art in the field of ultrasmall MRI contrast agents. The review focuses mostly on T1-contrast agents such as Gd complexes and small iron-based nanoparticles. Suitable for publication after revision. Article in a current state has several weaknesses that have to be addressed.

  1. Introduction: page 2, lines 73: the article is neglecting the role of other parameters such as coating and shape that importantly influence the contrast agent efficacy and their fate in the human body.
  2. Page 3: line 86: it is mention that particle size below a certain size limit can be efficiently excreted by the kidneys. However, surface charge plays a crucial role too. For example, despite the small size, negatively charged or neutral nanoparticles are more difficult to be filtered than positively charged counterparts. It is advised that this aspect is included in the discussion.
  3. The introduction should be rearranged to more clearly specify the advantages or disadvantages of Gd-based vs. iron-based contrast agents.
  4. Toxicity of Gd ions should be discussed more into details. For example, accumulation in the brain, bone, and kidney should be considered. How possible leakage of Gd ions from the complexes is evaluated in the described studies.
  5. Up to date list of clinically approved MRI contrast agents would be beneficial for the readers.
  6. What was the highest r1 achieved in the presented studies? If authors could discuss which factors influence the most r1 values, then the optimal design of contrast agents to achieve high r1 values could be proposed.
  7. Not only as T1 contrast agents, but manganese ions are used in T2 contrast agents as well. For example, due to the high magnetic moment, the addition of Mn to ferrites to improve magnetization and consequently r2.
  8. Page 15, line 500: r2/r1 ratio >5 is not low when we speak about T1 contrast agents. It is preferred to be in the range 1-2. For T2 contrast agents, this ratio should be above 10.

Author Response

For the Reviewer 1:

  • Introduction: page 2, lines 73: the article is neglecting the role of other parameters such as coating and shape that importantly influence the contrast agent efficacy and their fate in the human body.

Sentences ‘The contrast efficacy and in vivo fate of the MNP-based MRCAs are strongly dependent on their physical and chemical features including shape, size, surface charge, surface coating material and chemical/colloidal stability. For example, the PEGylated MNPs normally have relatively longer blood circulation time and higher colloidal/chemical stability than those of uncoated MNPs [11]. Very recently, the structure–relaxivity relationships of magnetic nanoparticles for MRI has been summarized in detail by Chen et al. [15].’ have been added in the revised manuscript for addressing this matter.

  • Page 3: line 86: it is mention that particle size below a certain size limit can be efficiently excreted by the kidneys. However, surface charge plays a crucial role too. For example, despite the small size, negatively charged or neutral nanoparticles are more difficult to be filtered than positively charged counterparts. It is advised that this aspect is included in the discussion.

A sentence ‘In addition, although they have similar small hydrodynamic diameters, the negatively charged or neutral NPs are more difficult to be eliminated by kidney than their positively charged counterparts.’ has been added in the revised manuscript for addressing this matter.

  • The introduction should be rearranged to more clearly specify the advantages or disadvantages of Gd-based vs. iron-based contrast agents.

A sentence ‘The advantages and disadvantages of Gd3+- and SPION-based MRCAs have been discussed in the reviews which published where else [3-19].’ has been added in the revised manuscript for addressing this matter .

  • Toxicity of Gd ions should be discussed more into details. For example, accumulation in the brain, bone, and kidney should be considered. How possible leakage of Gd ions from the complexes is evaluated in the described studies.

The safety issue of Gd3+ has been fully discussed in the ref. 9. A sentence ‘In particular, the advent of nephrogenic systemic fibrosis (NSF) and bone/brain deposition has led to increased regulatory scrutiny of the safety of the commercial Gd3+-chelates [9].’ has been added in the revised manuscript for addressing this matter.

  • Up to date list of clinically approved MRI contrast agents would be beneficial for the readers.

A table has been added in the revised manuscript for addressing this matter (Table 1).

  • What was the highest r1 achieved in the presented studies? If authors could discuss which factors influence the most r1 values, then the optimal design of contrast agents to achieve high r1 values could be proposed.

It is difficult to define the highest r1 values because of the variation of experimental conditions, in particular, the differences of MRI instrument parameters used in different experiments. Optimized the characteristics of MNDs could be generate MRCA with high performance, which have discussed in the section of ‘5 Conclusions and Outlook’.

  • Not only as T1 contrast agents, but manganese ions are used in T2 contrast agents as well. For example, due to the high magnetic moment, the addition of Mn to ferrites to improve magnetization and consequently r2.

Yes, it is indeed. We discussed the phenomenon in the section of ‘4 Dual paramagnetic metal nanodots’.

  • Page 15, line 500: r2/r1 ratio >5 is not low when we speak about T1 contrast agents. It is preferred to be in the range 1-2. For T2 contrast agents, this ratio should be above 10.

We are agree with the reviewer’s opinion. Here, the example was just used to show that the MR contrast capabilities of nanomaterials can be further improved while two paramagnetic metallic ions are integrated into one nanoplatform.

Reviewer 2 Report

The authors provide the state of the art of renal clearance of nanoparticle(NP)-based magnetic resonance imaging contrast agents.
The manuscript is well structured, starting with fundamental imaging modes of MRI and their applications, a brief summary of the properties of metals used for MRI probe design, and finally an overview of the design, (paramagnetic) characterization, and biological evaluation of nanoparticle-based MRI probes.
Comments and questions are below:
1) Authors can add a comment on the development of non-metal based MRI probes.
2) Biosafety of MRI probes are of paramount importance. Could the authors comment on what has been done to improve the design of classical MRI probes?
3) Is there an example where NP-based constructs show enhanced biosafety compared to classical MRI probe?
4) What are the main side effects of the metal leak from NP and where should these accumulate?
5) Several studies reviewed in the manuscript provide qualitative preclinical information. For the sake of consistency, available ID values should be provided for blood pool activity and renal clearance across all examples reported in the manuscript. 
6) Line 31-32: Is MRI radiation free?
7) Figure 7a can be improved.
8) line 418-419: "Using mouse model, the in vivo experiments..." the sentence should be more precise and keep consistency by stating the animal model and available ID values for blood pool activity and renal clearance.
line 456-457: The same information should be provided.

Author Response

Response to Reviewer 2

1 Authors can add a comment on the development of non-metal based MRI probes.

Yes, there are several non-metal based MRI probes, fluorine (F)-based materials. However, the non-metal based MRCAs are beyond the scope of this manuscript.

2 Biosafety of MRI probes are of paramount importance. Could the authors comment on what has been done to improve the design of classical MRI probes?

The safety issue of Gd3+ has been fully discussed in the ref. 9. Therefore, we didn’t discussed in our manuscript.

3 Is there an example where NP-based constructs show enhanced biosafety compared to classical MRI probe?

No, there are still no long-term toxicity comparison between the classical MRI probe and NP-based MRCAs.

4 What are the main side effects of the metal leak from NP and where should these accumulate?

For the large NPs, the metal leak from NPs may lead to liver injury or inflammation because these NPs have high accumulation amounts in the liver.

However, there are few studies on the in vivo toxicity of nanodots (i.e. NP with small size). It is believe that the biodistribution/toxicity of metal leak from nanodots is defined by characteristics of metal ion itself. We discussed the safety issue in the section of ‘5 Conclusions and Outlook’.

5 Several studies reviewed in the manuscript provide qualitative preclinical information. For the sake of consistency, available ID values should be provided for blood pool activity and renal clearance across all examples reported in the manuscript. 

The ID values were provided.

6 Line 31-32: Is MRI radiation free?

Yes, it is. See references 1 and 7.

7 Figure 7a can be improved.

It was improved in the revised version.

8 line 418-419: "Using mouse model, the in vivo experiments..." the sentence should be more precise and keep consistency by stating the animal model and available ID values for blood pool activity and renal clearance. line 456-457: The same information should be provided.

The mentioned information/values have been provided in the revised manuscript.

Round 2

Reviewer 1 Report

The authors have appropriately addressed all comments. Therefore, the paper is suitable for publication in its corrected form.